# The Association among Alcohol Consumption Patterns, Drink-Driving Behaviors, and the Harm from Alcohol-Related Road Traffic Injuries Due to the Drinking of Others in Thailand

**DOI:** 10.3390/ijerph192316281

**Published:** 2022-12-05

**Authors:** Sopit Nasueb, Jintana Jankhotkaew, Polathep Vichitkunakorn, Orratai Waleewong

**Affiliations:** 1International Health Policy Program, Ministry of Public Health, Tiwanon Road, Nonthaburi 11000, Thailand; 2Department of Family and Preventive Medicine, Faculty of Medicine, Prince of Songkla University, 15 Karnjanavanich Road, Songkhla 90110, Thailand

**Keywords:** road traffic injuries, drink driving, alcohol’s harm to others, Thailand

## Abstract

Thailand has one of the highest rates of traffic-related fatalities and alcohol-related road traffic injuries globally. Previous studies focused on alcohol consumption and road traffic injuries. However, no existing studies investigate the association between drink-driving behaviors and road traffic injuries due to the drinking of others. This study aims to explore any potential associations among alcohol drinking patterns, drink-driving behaviors, and the harm from alcohol-related road traffic injuries due to the drinking of others. The Thai Tobacco and Alcohol Use Household National Survey data in 2017 (*n* = 80,797) were analyzed using multiple logistic regression. This study found that current drinkers and binge drinkers were more likely to suffer from road traffic injuries due to others’ drink-driving behavior, i.e., 1.50 times (95% CI: 1.49–1.51) and 2.31 times (95% CI: 2.30–2.33), respectively, compared with non-drinkers. In addition, we found that drink-driving behavior was associated with harm from road traffic injuries due to others’ drink-driving behavior by 2.12 times (95% CI: 2.10–2.14) compared with the non-drinker group. This study calls for effective measures to reduce drink-driving behaviors to prevent road traffic injuries due to the drinking of others.

## 1. Introduction

Alcohol consumption is a potential risk factor for alcohol-related road traffic injuries. Road traffic injuries are among the leading causes of injury and death worldwide. The global status report on road safety shows an increase to 1.35 million deaths from road accidents worldwide in 2016, from 1.15 million in 2000, and a road traffic death rate of 18.2 people per 100,000, especially in low- or middle-income countries [1]. Thailand’s road traffic death rate of 32.7 people per 100,000 (or an average of 60 people per day) ranks ninth on the global stage and number one in Southeast Asia. Most deaths occur to motorcycle riders (74%) in the age group of 15–29 years, and 14% of all road traffic deaths are attributed to drinking and driving [1]. Road traffic injuries are the leading cause of death in Thailand [2].

Alcohol intoxication negatively affects cognitive and psychomotor abilities, and the risk of motor vehicle crashes increases dramatically when the driver consumes alcohol [3]. Several studies have investigated the association between alcohol consumption of those who suffered from road traffic injuries and describe the effect of drink-driving behaviors [4,5,6]. Studies from various countries confirmed that there is an association between alcohol consumption and injuries seen in the emergency room. For example, a meta-analysis of emergency room data confirmed that those who consumed more than 6 grams of alcohol on one occasion were two times more likely to experience injuries [7]. However, most of these studies did not specify those who suffered from injuries caused by their own drinking versus those from other’s drinking. Few studies have addressed pedestrians and passengers who suffered from drink-driving behaviors [8,9,10], irrespective of whether these individuals had consumed alcohol. A few studies investigated the alcohol consumption patterns of victims [9,11,12], but none of those investigated the alcohol consumption and drink-driving behaviors of the victims.

In low- and middle-income countries, drink-driving behaviors are still a major problem due to a lack of law enforcement caused by resource constraints [3]. However, no existing studies have investigated the association between drink-driving behaviors and experiencing harm from road traffic injuries that were caused by other’s drinking.

This study aims to explore the associations among drinking patterns, drink-driving behaviors, and alcohol’s harm to others from road traffic injuries in Thailand.

## 2. Materials and Methods

### 2.1. Data and Sample Population

We used a secondary database from the Thai Tobacco and Alcohol Use Household National Survey in 2017 conducted by the National Statistical Office (NSO). Participants were Thai and aged 11 years old and above. A two-stage stratified sampling method was applied. The survey was undertaken across 77 provinces in Thailand. The stratum for the sampling selection is a province. Each province was classified into two sub-strata: municipal and non-municipal areas. For each sub-stratum, enumeration areas (EA) were randomly selected. For each selected EA, households were randomly chosen. There were 2315 EAs selected from 129,440 EAs. Within the households, the NSO staff invited all members aged 11 years old and above to join the survey. The NSO staff collected data and all participants were informed by using official letters sent to the head of communities and participants before the face-to-face interviews (*n* = 120,003) [13]. Interviews were undertaken in Thai and translated into English for the article.

We included respondents aged 15 years old and above and reported the prevalence of alcohol consumption (*n* = 89,154). We excluded samples with missing data regarding the main independent variables (education: 225; marital status: 11; income: 7868; and occupation: 333). We also dropped respondents who were uncertain about the main dependent variable, i.e., those who reported not being sure whether the third parties drank alcohol, (*n* = 101). In total, we excluded the missing data of 8357 respondents; therefore, the final sample size was 80,797. The datasets had less than 10% missing data; therefore, a complete case analysis was performed [14].

### 2.2. Measurements

Characteristic variables included gender (male and female), age group (15–19, 20–24, 25–44, 45–59, and 60+ years), marital status (single, married, and divorced/widowed/separated), region (Bangkok, central, north, northeast, and south), area (rural and urban), and education (illiterate, primary, secondary, high school, and diploma or higher).

Socio-economic variables included employment status (unemployed and employed) and income status by quintiles (poorest, poor, middle, richer, and richest).

Alcohol consumption variables were determined by asking the respondents about their alcohol consumption behaviors via the two following questions:

(1) Have you ever drunk any liquor and alcoholic beverages? (“never drink alcohol in my lifetime”, “drank earlier but did not drink in the past 12 months”, and “drank in the past 12 months”, with the following frequencies: every day (7 days/week), almost every day (5–6 days/week), every other day (3–4 days/week), weekly (1–2 days/week), monthly (1–3 day/month), 8–11 days/year, 4–7 days/year, and 1–3 days/year).

(2) During the past 12 months, have you ever drunk heavily or engaged in binge drinking on at least one single occasion? (“never”, “ever drunk heavily in the past 12 months”, with the following frequencies: every day (7 days/week), almost every day (5–6 days/week), every other day (3–4 days/week), weekly (1–2 days/week), monthly (1–3 day/month), 8–11 days/year, 4–7 days/year, and 1–3 days/year). Binge-drinking is defined as the experience of drinking no less than 60 grams of pure alcohol on at least one single occasion or consuming with the following pattern:white liquor/herbal liquor: five shots or one fourth of a large bottle or half of a middle-sized bottle, ordistilled liquor: one fourth of a large bottle or five shots or eight glasses of distilled liquor containing a mixer, orbeer: four cans or two large bottles, orwine/champagne: one large bottle or four glasses of wine, orcider/wine coolers: four and a half bottles or cans, orfermented liquor (rice liquor/locally made liquor): one large bottle or two and a half glasses).

The respondents were provided pictures of alcoholic beverage products in order to assist respondents to visualize accordingly.

We categorized drinkers into abstainers (never drank during the past 12 months), and current drinkers (consuming alcohol in the previous 12 months). We also classified binge-drinking behaviors into three groups: abstainers (never drank during the past 12 months, non-binge drinkers (frequently drank but never experienced binge-drinking over the past 12 months), and binge-drinkers (experienced binge-drinking over the previous 12 months) [15].

Road traffic injury and drink-driving variables included road traffic injuries due to others’ drink-driving behavior, drink-driving, and harm from road traffic injuries due to self-drink-driving behaviors in the previous 12 months. The respondents were asked three questions.

(1)In the past 12 months, have you drunk liquor/alcohol beverages before or while driving a vehicle? There were four choices of answers, including (1) frequently, (2) sometimes, (3) driven but never drank before driving, and (4) never drove. This question was asked only among current drinkers. We classified choices 1 and 2 as drink-driving and choice 3 as non-drink-driving. We also excluded those who responded with choice 4 or non-drivers from the model that included drink-driving variables in order to avoid underestimation of drink-driving effect.(2)In the past 12 months, have you been injured or had an accident due to your own drinking before or while driving a vehicle? The choices of answers were never and frequently (one time, two times, three times, or more than three times). We classified those responded “frequently” as those who reported harm from road traffic injuries due to self-drink-driving behaviors.(3)In the past 12 months, have you been injured or had an accident by other people driving a vehicle? The choices of answers were never and frequently (i.e., was a passenger with intoxicated driver, was a passenger with unintoxicated driver but the opposite party was intoxicated, was a passenger with intoxicated driver and the opposite party was also intoxicated, was a passenger but was not sure whether the driver and/or the opposite party drank alcohol, and was a pedestrian (on the road/pavement) and was hit by intoxicated driver). We classified those who responded “frequently” as those who reported road traffic injuries due to others’ drink-driving behavior.

### 2.3. Data Analysis

Three main analytical approaches were applied in order to achieve the aims of this study. The first was to explore the prevalence of road traffic injuries due to other’s drinking behaviors, drink-driving behaviors, and harm from road traffic injuries due to self-drink-driving and to describe the demographic and socio-economic factors (Table 1). The second approach was to describe the distribution of populations across the three main outcomes: drink-driving, harms from road traffic injuries due to others’ drink-driving behavior, and harms from road traffic injuries due to self-drink-driving behavior (Figure 1).

The last approach was to examine the association among alcohol consumption patterns (prevalence of drinking and binge drinking), drink-driving behaviors, and harm from road traffic injuries due to others’ drink-driving behaviors (Table 2). Multiple logistic regression models were applied to determine the significant factors. We constructed four models using multiple logistic regression. The first model investigated an association between being current drinkers and harm from road traffic injuries due to others’ drink-driving behaviors (road traffic injuries (RTI) and harm to others (HTO)). The second model investigated an association between binge-drinking and RTI and HTO. The third model investigated an association between drink-driving behavior and RTI and HTO. This model excluded non-drivers. The last model investigated an association among drink-driving behavior and injuries experienced from their own drinking and RTI and HTO. This model excluded non-drivers. All analyses were performed using the STATA (version 14.2) program with statistical significance set at *p* < 0.05. The survey used complex multi-stage survey designs; therefore, sampling weights were applied in all descriptive and inferential analyses to account for sampling design, non-responses, and to allow the analysis to be nationally representative [13]. The NSO provided a weighted variable in order to adjust for complex survey design and to allow samples to be nationally representative. We used the variables adjusted for both descriptive and inferential analysis. NSO calculated the weighted variable using three main steps. First, design weight or base weight was calculated to estimate the reverse of probability of selected sample units at each stage of sample selection regarding survey design. Second, non-response rates were used to make adjustments. Third, a post-stratification calibration adjustment was conducted using the Thai population stratified by age, gender, province, region, and municipality to adjust the samples to be nationally representative. The weighted variable adjustments allowed the analysis in this study to be nationally representative [13]. In our study, the findings were based on the results of number of weighting factors.

### 2.4. Ethical Considerations

Ethical clearance for the secondary data analysis was given by the Human Research Ethics Committee of the Faculty of Medicine, Prince of Songkla University, Thailand (Ref No. 64-139-9-1).

## 3. Results

### 3.1. Demographic Characteristics, Socio-Economic Status, and Alcohol-Related Road Traffic Injuries

Among 50,460,216, one percent reported road traffic injuries due to others’ drink-driving behavior. The participants who reported a higher percentage of road traffic injuries due to others’ drink-driving behavior were male (1.35%). The study also found a higher percentage of road traffic injuries due to others’ drink-driving behavior among those aged 20–24 years (1.53%), single (1.17%), living in the northeast of Thailand (1.78%), living in rural environments (1.08%), and having graduated from high school (1.14%), compared with other groups. Furthermore, regarding socioeconomic status, it was found that 1.12% of employed persons and 1.17% of the middle-income group suffered from road traffic injuries due to other’s drink-driving behavior (Table 1).

The study found that those who reported a higher percentage of drink-driving behaviors were males (58.70%), aged 15–19 years (66.78%), single (60.15%), living in the north region (67.47%), and living in rural areas (59.27%), compared with other groups. Regarding socio-economic status, 56.02% of working people and 61.17% of the middle-income group reported drink-driving behavior (Table 1).

In addition, the study also found that participants who reported a higher percentage of harm from road traffic injuries due to self-drink-driving behavior were males (5.50%) and aged between 15–19 years (7.12%), compared with other groups. Regarding socio-economic status, 6.89% of the participants were unemployed and 9.05% of the poorest group suffered from road traffic injuries due to self-drink-driving behavior. See more details of the distribution of both samples and population in Appendix A.

### 3.2. Self-Reported Drinking Behavior and Experience of Road Traffic Injury Due to Others’ Drink-Driving Behavior

The study found that the prevalence of current drinkers among Thais was approximately 29.82%, and the prevalence of binge-drinkers was approximately 12.57%. Regarding the drinking behavior among the group who had been injured from road traffic injuries due to others’ drink-driving behavior, 45.63% had drunk in the past 12 months and 16.04% had experienced binge-drinking.

Figure 1 outlines multiple experiences of risky behavior and alcohol-related harms, including drink-driving behaviors, harm from road traffic injuries due to others’ drink-driving behavior, and harm from road traffic injuries due to self-drink-driving behavior. Among those who reported one of those harms and drink-driving behaviors (6,250,007), 39,776 people had three experiences: drink-driving behaviors, harm from road traffic injuries due to others’ drink-driving behavior, and harm from road traffic injuries due to self-drink-driving behavior. In addition, 107,852 people had a joint experience of drink-driving and harm from road traffic injuries due to others’ drink-driving behavior. Furthermore, 5,738,452 people had only one experience of drink-driving but were never injured. In addition, 377,959 people had an experience of road traffic injuries due to others’ drink-driving behavior but never committed drink-driving behavior, and they never experienced harm from road traffic injuries from self-drink-driving. Furthermore, 271,714 people were injured in connection with their drinking behavior, but never experienced harm from road traffic injuries from other’s drink-driving behavior.

### 3.3. Factors Associated with Road Traffic-Related Injuries and Ones’ Own Alcohol Drinking Behavior

A multivariable logistic regression yielded the finding that current drinkers were more likely to suffer from road traffic injuries due to others’ drink-driving behavior by 1.50 times (95% CI: 1.49 to 1.51) vs. non-drinkers. In addition, we found that binge-drinkers were more likely to experience harm from road traffic injuries due to others’ drink-driving behavior by 2.31 times (95% CI: 2.30 to 2.33) vs. non-drinkers. However, those who were current drinkers but were not binge drinkers over the past 12 months were less likely to experience harm compared with non-drinkers. However, the effect was minimal, with an odds ratio close to 1. In addition, we found that respondents who committed drink-driving were more likely to suffer from road traffic injuries due to others’ drink-driving behavior by 2.12 times (95% CI: 2.10 to 2.14) vs. non-drinkers. Furthermore, we found that those who frequently committed drink-driving and experienced injury from their own drinking were more likely to suffer from road traffic injuries due to the drink-driving of others by 11.57 times (95% CI: 11.43 to 11.71) vs. non-drinkers (Table 2).

In addition, we also found associations between all independent variables and harm from road traffic injuries due to the drinking of others.

## 4. Discussion

The main findings from this study showed that current drinking and binge drinking were associated with harms from road traffic injuries due to others’ drink-driving behavior, and a person’s drink-driving behavior increases their risk for road-traffic injuries due to other’ drink-driving behaviors.

Those who were drinkers and binge-drinkers were more likely to suffer from road traffic injuries due to the drinking of others. The findings of this study offered two main contributions. First, our study contributed to identifying the determinants of road traffic injuries by identifying the risk groups who suffer from road traffic injuries due to drinking of others, including drinking and binge-drinking. This goes beyond the knowledge of injuries seen in emergency rooms which identifies alcohol consumption mainly of those who were injured [7]. These findings also echo that reducing alcohol consumption can not only prevent self-harm but also harm to other people. Second, these findings confirmed previous studies that focused on alcohol’s harm to others [10,16,17]. For example, a study from Thailand found that those who were drinkers (regular and occasional drinkers) and binge drinkers were more likely to experience alcohol-related harm from others (social harm and psychological harm) than abstainers [10]. This is similar to studies conducted in high-income countries such as Denmark, Norway, and Sweden. [16,17].

This study found that drink-drivers were two times more likely to experience road traffic injuries due to the drinking of others compared with non-drinkers. This is the first study to set a precedent to discover that drink-driving is a risk of road traffic injury due to others’ drinking. A previous study in New Zealand found that more than 40% of people who suffered from road traffic injuries had not themselves been drinking [18]. This study investigated beyond drinking patterns by examining drink-driving, which is a great challenge in low- and middle-income countries [3]. Furthermore, it discovered that those who drink and drive were two times more likely to suffer from road traffic injuries due to the drinking of others. This was not a surprise, as the cognitive and psychomotor abilities of respondents who drink and drive would typically be impaired; therefore, they would be prone to receive the highest level of impact due to drink-driving behavior of others [19]. Another explanation may be that those who drink and drive do this more often at particular times of the day and week, i.e., during weekend nights when more drunk drivers are using the road. Therefore, it is more likely that a drunk driver causes a crash in which the other road user is also a current drinker or a binge-drinker.

### 4.1. Strengths and Limitations

The main strength of this study was the very large number of surveyed participants, which is a nationally representative survey. However, there were some limitations. First, this study was based on a cross-sectional design; therefore, a causal relationship cannot be confirmed. Second, this study may be underreporting road traffic injuries because of rare cases of those who experienced road traffic injuries due to others’ drinking. Therefore, to have a comprehensive picture of road traffic injuries due to other’s drinking, a more effective study design would be to also use registry data (e.g., hospital-based data or police data), which often report road traffic injuries, in combination with surveys. Third, respondents may be reluctant to report drink-driving behaviors and harms due to their own drink-driving behaviors; therefore, this study may suffer from respondents underreporting the harms from road traffic injuries due to self-drink-driving behaviors. In addition, the respondents’ reporting harms from road traffic injuries due to other’s drink-driving behaviors is largely based on their subjective views; therefore, this study may have had an over- or underreporting of harms. Furthermore, this study was a quantitative case of research, and it was potentially insufficient regarding properly understanding the context. Thus, qualitative research may be more suitable, especially with regard to the impact of harm from other’s drink-driving behavior. Last, we acknowledged that the survey applied a two-stage sampling method by randomly selecting enumeration areas and households. All members within the household were interviewed; therefore, there might be the case that there is a cluster within the household. However, as the sampling weight is already adjusted for individual and household samples to be nationally representative, the problem of clustering was possibly minimized.

### 4.2. Policy Implications

We found that drink drivers caused harm to themselves and are likely to suffer harm due to other people’s drinking. Therefore, the implementation of policies or measures must incorporate various strategies for the management of the problem of drink-driving, such as laws that lower blood alcohol concentration limits for younger drivers, random breath testing, and public awareness campaigns to improve attitudes among people. We recommend that the government’s policy on drink-driving should be strictly enforced.

## 5. Conclusions

Based on our study, it is seen that being a current drinker and binge-drinker increased the likelihood of experiencing harm from road traffic injuries due to the drinking of others. In addition, drink-driving behaviors also increase the risk of suffering from road traffic injuries due to the drinking of others. The study echoed that in order to prevent road traffic injuries, effective measures and effective implementation to reduce alcohol consumption as well as drink-driving behaviors are urgently needed. It is urgently required that the existing laws be effectively enforced to prevent drink-driving since Thailand’s leading cause of death is due to road traffic injuries.

## Figures and Tables

**Figure 1 ijerph-19-16281-f001:**
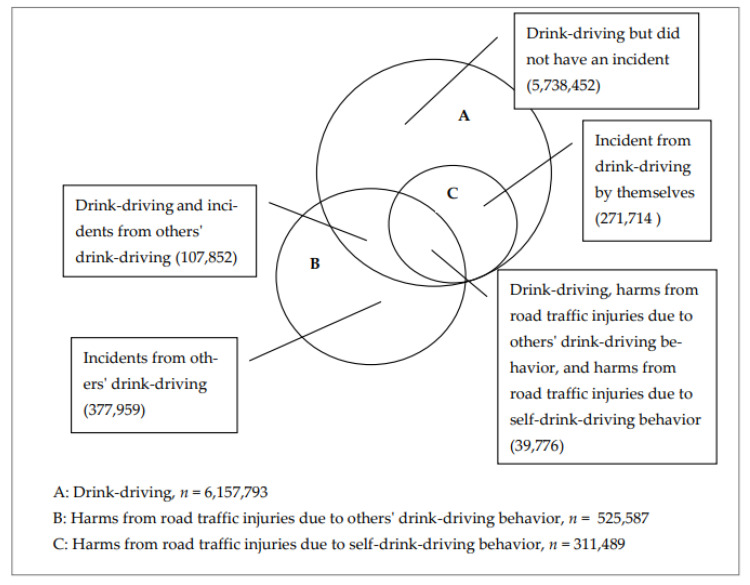
Proportion of reported drink-driving behavior and harms from road traffic injuries due to self- and others’ drink-driving behavior.

**Table 1 ijerph-19-16281-t001:** Descriptive results of demographical characteristics, socio-economic, drink-driving behavior, and harm from road traffic injuries due to self-drink-driving and others’ drink-driving behavior.

Variable	Road Traffic Injuries Due to Others’ Drink-Driving Behavior			Drink-Driving			Harms from Road Traffic Injuries Due to Self-Drink-Driving Behavior		
	Total	No	Yes	Total	No	Yes	Total	No	Yes
	*n* = 50,460,216	*n* = 49,934,630	*n* = 525,587	*n* = 11,028,449	*n* = 4,870,656	*n* = 6,157,793	*n* = 6,157,793	*n* = 5,846,304	*n* = 311,489
**Demographic Characteristics**									
Gender									
Female	25,337,036	99.27	0.73	1,669,219	60.25	39.75	663,529	98.58	1.42
Male	25,123,180	98.65	1.35	9,359,230	41.30	58.70	5,494,264	94.50	5.50
Age group									
15–19 years	3,188,928	98.92	1.08	415,342	33.22	66.78	277,381	92.88	7.12
20–24 years	3,983,154	98.47	1.53	1,114,648	37.69	62.31	694,562	94.58	5.42
25–44 years	18,282,069	98.98	1.02	5,219,448	44.88	55.12	2,876,899	95.19	4.81
45–59 years	13,979,323	98.98	1.02	3,353,441	45.04	54.96	1,842,933	95.16	4.84
60+ years	11,026,742	99.09	0.91	925,570	49.65	50.35	466,018	94.34	5.66
Marital status									
Single	11,908,964	98.83	1.17	2,663,421	39.85	60.15	1,602,018	94.03	5.97
Married	31,882,292	98.98	1.02	7,525,616	45.96	54.04	4,066,941	95.38	4.62
Divorced/widowed/separated	6,668,961	99.09	0.91	839,412	45.96	54.04	488,834	94.27	5.73
Region									
Bangkok	7,082,856	99.46	0.54	1,126,807	64.93	35.07	395,222	95.19	4.81
Central	14,854,799	99.18	0.82	3,172,182	49.89	50.11	1,589,682	94.96	5.04
North	8,628,601	99.13	0.87	2,447,347	32.53	67.47	1,651,261	96.82	3.18
Northeast	13,344,282	98.22	1.78	3,426,680	42.49	57.51	1,970,541	93.62	6.38
South	6,549,678	99.21	0.79	855,433	35.58	64.42	551,087	93.82	6.18
Area									
Rural	27,313,442	98.92	1.08	6,368,201	40.73	59.27	3,774,634	94.94	5.06
Urban	23,146,774	99.00	1.00	4,660,248	48.86	51.14	2,383,159	94.94	5.06
Education									
Illiterate	2,361,988	99.26	0.74	195,144	49.84	50.16	97,884	92.64	7.36
Primary school	22,899,870	98.89	1.11	4,442,968	41.78	58.22	2,586,780	93.58	6.42
Secondary school	8,383,518	98.88	1.12	2,207,363	42.49	57.51	1,269,560	94.72	5.28
High school	7,981,081	98.86	1.14	2,217,553	43.92	56.08	1,243,691	95.75	4.25
Diploma or higher	8,833,759	99.22	0.78	1,965,422	51.16	48.84	959,878	98.09	1.91
**Socio-Economic Status**									
Employment status									
Unemployed	13,058,323	99.19	0.81	902,748	46.22	53.78	485,462	93.11	6.89
Employed	37,401,894	98.88	1.12	|10,125,701	43.98	56.02	5,672,331	95.10	4.90
Monthly (individual) income									
Poorest (quintile 1)	9,017,346	98.92	1.08	860,929	43.28	56.72	488,361	90.95	9.05
Poor (quintile 2)	11,102,039	98.90	1.10	2,006,246	39.22	60.78	1,219,418	95.25	4.75
Middle (quintile 3)	8,036,462	98.83	1.17	2,192,836	38.83	61.17	1,341,252	94.35	5.65
Richer (quintile 4)	10,443,773	99.02	0.98	2,759,977	43.88	56.12	1,548,857	94.47	5.53
Richest (quintile 5)	11,860,596	99.07	0.93	3,208,462	51.38	48.62	1,559,904	96.93	3.07

**Table 2 ijerph-19-16281-t002:** Factors associated with the respondents’ current drinking, binge drinking, drink-driving behavior, harm from road traffic injuries due to self-drink-driving behavior, and having experienced harm from road traffic injuries due to others’ drink-driving behavior (*n* = 80,893).

Variable	Model 1: RTI and HTO *n* = 80,797		Model 2: RTI and HTO *n* = 80,797		Model 3: RTI and HTO *n* = 75,305		Model 4: RTI and HTO *n* = 75,305	
	AOR	95% CI	AOR	95% CI	AOR	95% CI	AOR	95% CI
Current drinking in past 12 months								
Non-drinker	1							
Drink	1.50 **	(1.49–1.51)						
Binge-drinking								
Non-drinker			1					
Drink without binging			0.97 **	(0.97–0.98)				
Binge-drinker			2.31 **	(2.30–2.33)				
Drink and driving								
Non-drinker					1			
Drink without driving					1.20 **	(1.19–1.21)		
Drink-driving					2.12 **	(2.10–2.14)		
Drink-driving and injury								
Non-drinker							1	
Drink without driving							1.20 **	(1.99–1.21)
Drink-drive and no injury from self-drinking							1.64 **	(1.62–1.65)
Drink-drive and injury from self-drinking							11.57 **	(11.43–11.71)
**Demographic Characteristics**								
Gender								
Female	1		1		1		1	
Male	1.53 **	(1.52–1.54)	1.44 **	(1.43–1.45)	1.60 **	(1.59–1.61)	1.57 **	(1.56–1.58)
Age group								
15–19 years	1		1		1		1	
20–24 years	1.35 **	(1.33–1.37)	1.27 **	(1.25–1.29)	1.38 **	(1.36–1.40)	1.36 **	(1.34–1.38)
25–44 years	0.98 **	(0.96–0.99)	0.93 **	(0.92–0.94)	0.98	(0.97–1.00)	0.97 **	(0.86–0.99)
45–59 years	0.90 **	(0.88–0.91)	0.87 **	(0.86–0.88)	0.90 **	(0.88- 0.91)	0.89 **	(0.88–0.91)
60+ years	0.89 **	(0.88–0.90)	0.89 **	(0.88–0.90)	0.95 **	(0.94–0.97)	0.96 **	(0.95–0.98)
Marital status								
Single	1		1		1		1	
Married	0.87 **	(0.86–0.87)	0.89 **	(0.88–0.90)	0.87 **	(0.86–0.88)	0.89 **	(0.88–0.90)
Divorced/widowed/separated	0.95 **	(0.94–0.96)	0.94 **	(0.93–0.95)	0.96 **	(0.96–0.98)	0.98 **	(0.97–1.00)
Region								
Bangkok	1		1		1		1	
Central	1.65 **	(1.64–1.68)	1.65 **	(1.63–1.67)	1.90 **	(1.88–1.93)	1.91 **	(1.89–1.94)
North	1.74 **	(1.72–1.76)	1,73 **	(1.71–1.75)	1.89 **	(1.86–1.91)	1.96 **	(1.93–1.99)
Northeast	3.59 **	(3.54–3.63)	3.53 **	(3.49–3.57)	3.94 **	(3.89–3.99)	3.92 **	(3.87–3.94)
South	1.71 **	(1.69–1.73)	1.72 **	(1.69–1.74)	1.82 **	(1.58–1.85)	1.81 **	(1.79–1.84)
Area								
Rural	1		1		1		1	
Urban	1.28 **	(1.27–1.28)	1.28 **	(1.28–1.29)	1.32 **	(1.32–1.33)	1.30 **	(1.30–1.31)
Education								
Illiterate	1		1		1		1	
Primary school	1.13 **	(1.12–1.15)	1.13 **	(1.11–1.15)	0.99	(0.97–1.01)	0.99	(0.98–1.01)
Secondary school	1.04 **	(1.02–1.06)	1.02 *	(1.00–1.04)	0.86 **	(0.85–0.88)	0.88 **	(0.86–0.89)
High school	1.08 **	(1.06–1.10)	1.08 **	(1.06–1.10)	0.94 **	(0.93–0.96)	0.98 **	(0.97–1.00)
Diploma or higher	0.85 **	(0.83–0.86)	0.86 **	(0.84–0.87)	0.77 **	(0.76–0.79)	0.82 **	(0.80–0.83)
**Socio-Economic Status**								
Employment status								
Unemployed	1		1		1		1	
Employed	0.89 **	(1.28–1.30)	1.28 **	(1.27–1.29)	1.28 **	(1.27–1.29)	1.30 **	(1.28–1.31)
Monthly (individual) Income								
Poorest (quintile 1)	1		1		1.00			
Poor (quintile 2)	0.89 **	(0.88–0.90)	0.89 **	(0.88- 0.90)	0.99 **	(0.89–0.91)	0.92 **	(0.91–0.93)
Middle (quintile 3)	0.85 **	(0.84–0.85)	0.85 **	(0.84–0.86)	0.84 **	(0.83–0.84)	0.85 **	(0.84–0.86)
Richer (quintile 4)	0.79 **	(0.78–0.80)	0.80 **	(0.79–0.81)	0.71 **	(0.70–0.72)	0.73 **	(0.72–0.74)
Richest (quintile 5)	0.93 **	(0.92–0.94)	0.93 **	(0.92–0.94)	0.92 **	(0.91–0.93)	0.95 **	(0.94–0.97)

RTI and HTO represent having experienced harm from road traffic injuries due to others’ drink-driving behavior. Model 1 investigated an association between being a current drinker and RTI and HTO. Model 2 investigated an association between binge-drinking and RTI and HTO. Model 3 investigated an association between drink-driving behavior and RTI and HTO. This model excluded non-drivers. Model 4 investigated an association among drink-driving behavior and experienced injuries from their own drinking and RTI and HTO. This model also excluded non-drivers. AOR is the adjusted odds ratio. * *p* < 0.05 ** *p* < 0.01.

## Data Availability

Data and materials will be provided by the corresponding author upon request.

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
