# Peer review of "The Association among Alcohol Consumption Patterns, Drink-Driving Behaviors, and the Harm from Alcohol-Related Road Traffic Injuries Due to the Drinking of Others in Thailand"

_ijerph, 2022, doi:10.3390/ijerph192316281_

Round 1
Reviewer 1 Report (Previous Reviewer 1)
All my comments have been addressed adequately. I have no new comments except that probably due to the conversion to pdf, text is missing in the text boxes of Figure 1.
Author Response
Dear reviewer,
We adjust figure 1 as per your suggestion.
Best regards,
Jintana
This manuscript is a resubmission of an earlier submission. The following is a list of the peer review reports and author responses from that submission.
Round 1
Reviewer 1 Report
The researchers wanted to know if the risk of getting injured in a traffic crash by a drunk driver or rider is associated with the victim’s own drinking and drink-driving patterns. To this end they analyzed the data of the Thai Tobacco and Alcohol Use Household National Survey of 2017. 80,893 respondents were included. Of these respondents 882 (1.09%) mentioned that they got injured in a road traffic crash within the past 12 months in which the other road user who caused the crash, was drunk. They conducted a logistic regression with ‘being injured due to other’s drink-driving behavior in the previous 12 months’ (yes/no) as the dependent variable and demographics (gender, age group, marital status, region, education level, employment status, income status), own drinking patterns (has not consumed alcohol in the past 12 months, has at least one time consumed alcohol in the past 12 months (a current drinker), has at least consumed one time in the past 12 months more than 60 grams of alcohol during a single occasion (a binge drinker)), and having driven at least one time under the influence of alcohol in the past 12 months as predictors. They found that ‘current drinkers’ and even more so ‘binge drinkers’ were more likely to get injured in a traffic crash caused by a drunk driver than road users who do not consume alcohol, adjusted OR (i.e. an OR in which is controlled for the other predictors in the model) of 1.53 (95% CI: 1.31 – 1.79) and of 2.49 (95% CI: 2.07-2.99) respectively. Road users who had stated they had driven under the influence of alcohol in the past 12 months had an adjusted OR of 2.37 (95% CI: 1.98 – 2.85) compared to those who had stated not to have driven under the influence of alcohol. Furthermore, males were more at risk of getting injured due to a drunk road user than females, adjusted OR is 1.40 (95% CI: 1.30-1.64). The risk of getting injured by a drunk road user was higher in some regions of Thailand than others and was higher in urban areas. The other demographic variables such as age group, marital status, income, and education level did not affect the risk of getting injured by a drunk road user.
The article is concise, well written and well organized. Furthermore, the applied method is simple but accurate. I have one more general comment some detailed comments that to my opinion are caused by the fact that the article is somewhat too concise.
General comments
The authors do not provide the exact questions that were posed in the survey. What for instance was the definition of being injured as a result of a traffic crash? Was being injured for instance having an injury on the Maximum Abbreviated Injury Scale (MAIS) of 3 or more? When it is not accurately defined some respondent may consider themselves as injured whereas others with the same injury consider themselves as not being injured. Furthermore, what was the definition of a traffic crash. As single vehicle crashes were excluded because there had to be a collision partner that could be under the influence of alcohol or not that ‘caused’ the crash, was this a collision with another road user on public roads?
How could the respondents be sure that the other road user in the crash ‘caused’ the crash? Traffic crashes have most of the times multiple (underlying) causes and people tend to blame the other and do not mention their own errors, mistakes, or violations as the causes of a crash. It could be that they state that the other caused the crash whereas in fact they themselves caused it. This is not mentioned as a limitation of the study.
Most importantly, how could the victim know that the collision partner was under the influence of alcohol? Was this always the police who had measured the blood alcohol level of the other road user and had told the victim that the other road user was drunk? Or did the victim only suspect that the other road user was under the influence of alcohol? When it is the latter, the number of crashes due the drunk driving of others may be overestimated. This is not mentioned as a limitation.
Detailed comments
Abstract (and section 2.2. (Measurement)). In the abstract the authors speak about ‘current drinkers’ and ‘binge drinkers’ without defining what a current drinker and a binge drinker is. According to what is mentioned in section 2.2 a current drinker is ‘experience of alcohol drinking in the previous 12 months’ and binge drinker is ‘experience of drinking no less than 60 grams of pure alcohol on at least one single occasion in the previous 12 months.’ This implies that someone who had only once a year consumed one standard glass of alcohol is a ‘current drinker’ and someone who has only one time in year has consumed 6 standard glazes of alcohol on one occasion (for instance at a wedding), is a binge drinker. To me someone who only once a year has consumed 6 standard glasses of alcohol is not a binge drinker. No references are provided in section 2.2. for the definition of a current drinker and a binge drinker. Please refer in section 2.2 to existing definitions of a current drinker and a binge drinker.
Section 2.1 (Data and Sample Population). The authors state that multi-stage stratified sampling from 77 provinces in Thailand. Does this imply that the sample matched the number of inhabitants in each of the Thai provinces? Furthermore, in section 4.1. (Strength and Limitation) the authors speak about a ‘very large number of surveyed participants and weighing sample’. Is this weighing sample the same as the stratified sample or was a weigh factor applied to make the sample representative for road users in Thailand? For instance, 52.8% of the respondents were female and 47.2% were male. I think that as in most countries there are approximately as many females as males. Did the authors applied a weigh factor for gender?
Section 2.2. (Measurement). In this section definitions are provided for a ‘current drinker and a ‘binge drinker’ (see my previous comment0 but not for a ‘drink-driver’. What was the question that measured drink-driving? Was this someone who had at least once in the past 12 months driven under the influence of alcohol?
Figure 1. Part of the last line in this figure is missing.
Section 4 (Discussion). Last sentence before section 4.1. (Strength and Limitation). The authors write; ‘This is not surprise, ….’ I think it is better to write ‘This is not a surprise’ or ‘This is not surprising’.
Section 4 (Discussion). The same last sentence. The authors write: “This is not surprise, when respondents who drink and drive would affect cognitive and psychomotor abilities; therefore, they prone to have the most effect from drink-driving behavior of others.” This may indeed be a possible explanation. Another may be that those who drink and drive do this more often at particular times of the day and particular times of the week, that is during weekend nights. In periods with more drunk road users it is more likely that a drunk driver causes a crash in which the other road user is a current drinker or a binge drinker.
Section 4.1. (Strength and Limitation). The authors write: “Therefore, to have comprehensive picture of road traffic injuries due to other’s drinking, the study in setting where road traffic injuries reported should be combined.” I do not understand this sentence.
Author Response
Dear Reviewers,
Thank you very much for your comments. We revised the paper in response to the comments, your comments will be a valuable contribution to the literature. Please see attached file for an explanation of the reviewer’s comment in more detail.
Best regards,
Sopit
On behalf of all authors

Reviewer 2 Report
The Author(s) conducted an interesting study on the effect of alcohol on traffic injuries suffered by drink-drivers because of other drinker-drivers or self-drink-driving.
Data from a Thai Tobacco and Alcohol Use House-hold National database (2017) on a wide and adequately stratified sample (age, gender, region and area of origin, marital status and education, employment and income status) were analyzed to provide a contribution to the understanding of the effect of alcohol use on the risk to be involved in accidents caused by oneself or by others. Participants were current- and binge-drinkers in the previous 12 months and were administered a face-to-face interview.
Methodology and descriptive statistics are well illustrated. Results of statistical analyses are discussed. Conclusions seem supported by the results. Tables are adequate (see minor points for an incoherence between what reported in Table 1 and Figure 1).
Minor points:
- In section 3.1, line sixth, “living in the south of Thailand (1.18%)” should be replaced with “living in the northeast of Thailand (1.75%)”.
- In Table 1 the number of Drink-driving (9,413) does not correspond with that indicated in Figure 1 (9,431). Please, correct the typo.
- Figure 1: some lines are not clearly visible, both in the text-box included in the figure and in the legend below.
- Discussion: the distinction between the first and the second conclusion (first 4 lines) is not so clear and the same lack of clarity appears in lines three to six of the Conclusion section. These sentences should be rephrased.
Author Response

(The authors gave the same response as above.)
